# Enhancing childhood immunization coverage in Mozambique and Malawi: Study protocol of a mixed methods evaluation of the 'Let's talk about vaccines' multisite community-based participatory project

**Linda Shuro** [1]*, **Emily Lawrence**[2], **Lucia Knight**[3], **Helen Schneider**[4], **Hanani Tabana**[1]

**1** School of Public Health, Faculty of Community and Health Sciences, University of the Western Cape, Cape Town, South Africa, **2** VillageReach, Seattle, Washington, United States of America, **3** School of Public Health & Family Medicine, Faculty of Health Sciences, University of Cape Town, Cape Town, South Africa, **4** School of Public Health & SAMRC Health Services to Systems Research Unit, Faculty of Community and Health Sciences, University of the Western Cape, Cape Town, South Africa

* 2948252@myuwc.ac.za

## Abstract

### Background

Full coverage of childhood vaccines is a persistent challenge in low- and middle-income countries, suggesting the presence of specific contextual barriers. The emergence of the COVID-19 pandemic further worsened the situation. The complementary use of community-based participatory research (CBPR) and human-centered design (HCD) approaches has the potential to effectively create tailored interventions for improving public health outcomes. This protocol provides examples of methods to evaluate the use of these novel approaches in low- and middle-income countries. The 'Let's talk about vaccines' project is a multisite community-based participatory project by VillageReach that uses the CBPR and HCD approaches to identify the barriers and co-create interventions/solutions to address under two routine immunization access and uptake in Malawi and Mozambique.

### Methods

Guided by the Reach Effectiveness Adoption Implementation Maintenance (RE-AIM) framework, this evaluation prospectively evaluates the effectiveness (on under-two immunization coverage and related outcomes) of VillageReach's co-created interventions and processes of implementation (reach, adoption, implementation and maintenance) in two districts in Mozambique and two in Malawi. This paper will also describe the theory of change for VillageReach's project. Thematic analysis will be used to analyze the qualitative data, and interrupted time series analysis used to analyze the co-created interventions' effectiveness on specific under two immunization outcomes. The analysis will integrate complex systems thinking and constructs inherent in health systems strengthening.

**Data Availability Statement:** No datasets were generated or analysed during the current study. All relevant data from this study will be made available upon study completion. Deidentified research data will be made publicly available when the study is completed and published. To maintain confidentiality, demographic data will be aggregated as much as possible. All participants' data will be allocated pseudo-IDs. These IDs will be used for data entry and analysis. Any connections between participants' names and pseudo-IDs will be kept in password protected electronic file on a secure server on a password protected computer. All paper data will be locked in a file cabinet accessible to the evaluation team only. A data sharing agreement will be put in place and all implementing partners and funders will be trained and will be responsible for adhering to the data management process. At the end of the evaluation and publication processes, the data will be archived, and the evaluation team will serve as the steward for the de-identified dataset. For future analysis the paper data will be kept for 5 years and 15 years for the electronic data. Thereafter the paper data will be shredded and the electronic will be encrypted and deleted.

**Funding:** Initial of authors who received award: E.L Funder: Wellcome Trust https://wellcome.org/ Wellcome Trust had no role in the design of this study. The funding agency will also not play any role in the execution of the study, analysis and interpretation of the data.

**Competing interests:** The authors declare that they have no competing interests.

## Discussion

This evaluation is an opportunity to share the use of novel and best practices, opportunities and challenges for improved community-responsive programming in routine immunization. It will be fundamental in providing evidence on the impact of interventions, evidence on mechanisms behind improvements in under- two immunization outcomes due to code-signed community-driven solutions and informing their scalability in similar contexts. Findings will inform the development of a comprehensive framework to guide the scalability of community-based approaches on childhood immunization uptake and access into similar contexts.

## Introduction

Vaccination coverage is a critical measure of immunization performance, representing the percentage of children within a specific area who receive the recommended vaccines among a target population [1]. Childhood vaccinations represent one of the most cost-effective public health interventions [2, 3], averting an estimated 4.4 million deaths annually [4, 5]. Despite their proven effectiveness and substantial investments, full childhood immunization coverage reached only 86% in 2019, below the World Health Organization's (WHO) target of 95% [6]. The COVID-19 pandemic exacerbated the situation, with an estimated 67 million children missing out on routine immunizations between 2019 and 2021 [7]. Compared to pre-pandemic levels in 2019, recent data released by WHO and United Nations International Children's Emergency Fund (UNICEF) indicate that childhood immunization coverage stagnated in 2023, resulting in an additional 2.7 million children unvaccinated or under-vaccinated [8]. In Mozambique and Malawi, while initial vaccine doses have high coverage, there is a notable drop in the completion rates of vaccine series [9]. This drop suggests the presence of specific barriers. Some studies highlight that demand-side factors such as vaccine hesitancy, mistrust in the health care system, pockets of religious sectors and related beliefs against immunization, misinformation and lack of knowledge on immunization play a significant role in dropout rates. Other studies emphasize that supply-side factors such as logistical challenges and limited access to health care are significant in hindering immunization efforts [9–11]. These factors vary in low and middle-income countries (LMICs) and globally, and are highly contextual, varying from individual, interpersonal, and system-related factors [12, 13]. Understanding these barriers is crucial in developing targeted strategies. To catch up on the children left behind during the pandemic as well as reach those that public health systems have persistently missed, global agencies such as UNICEF, the Global Alliance for Vaccine and Immunization (GAVI) and the WHO have called for building trust and demand for vaccines within communities and addressing critical gaps and obstacles to restoring routine immunization [14].

Community-based participatory research (CBPR) and human-centered design (HCD) are emerging as promising methodologies for designing tailored public health interventions that improve health outcomes and service trust [15–18]. These approaches are particularly useful among marginalized communities, as they create collaborative relationships among groups and shared control over health and social problems [19–21]. CBPR is an inclusive and collaborative approach to research that involves the researchers and community stakeholders in the research process with shared power between researchers and participants, recognition of experiential understanding, a focus on improvement in circumstances and

implementation [19, 20]. HCD integrates an inclusive participatory process that results in the collaborative development of solutions to problems, buy-in by stakeholders and better-tailored solutions. It is a community-driven approach that centers on creativity in an iterative process in the development of feasible solutions. It consists of three main phases: inspiration, ideation and implementation [22]. In particular, these approaches may play a crucial role in trying to improve full immunization coverage, as new solutions to improve vaccination coverage have been primarily driven by international stakeholders and national government decision-makers [23], while caregivers and health care workers who directly interact with the community have not been engaged in identifying barriers and solutions to address them [23–27].

The hypothesis posits that interventions co-created through CBPR and HCD will potentially increase the rate of complete childhood immunizations in targeted districts of Mozambique and Malawi compared to baseline and control sites. Moreover, increased engagement and trust among caregivers and health workers, fostered by CBPR and HCD, correlate with higher immunization uptake and completion rates.

## 'Let's talk about vaccines' project (the Project)

In an attempt to improve under two immunization coverage and catch up on children who have missed out on vaccines, VillageReach is implementing the 'Let's talk about vaccines' project, 'Bate-Papo' in Portuguese, henceforth referred to as "the Project". The Project aims to identify and address barriers to routine immunization dropouts in two districts in Mozambique and two districts in Malawi [12, 13, 28]. It seeks to amplify caregiver and health worker voices because they know best what barriers they face and how to address them. The Project uses principles of CBPR and HCD to generate new knowledge and targeted solutions that meet their needs. Co-created interventions are expected to be context-specific addressing the identified barriers. These may include interventions to improve immunization knowledge through culturally relevant information education and communication material, demystify myths and misconceptions, reduce vaccine hesitancy, and collaborative interventions or tools to improve trust, programming, and access to immunization [18, 29, 30]. While there is a growing, yet limited, body of research in Africa that robustly applied HCD to improve public health outcomes, these studies are mostly in HIV, non-communicable diseases, and reproductive health [31–34]. There are very few studies on improving community-responsive programming in childhood immunization. VillageReach will pilot the co-created interventions to identify best practices for engaging caregivers and health service providers to improve routine immunization and understand how to replicate and scale this approach and the solutions to other contexts. VillageReach is partnering with the University of the Western Cape and University of Cape Town to evaluate the Project.

## Evaluation of the Project

There is a dearth of evaluation studies, especially in LMICs, that provide evidence on the evaluation of interventions co-created through CBPR and HCD, particularly within the immunization space [18, 35]. By documenting and analyzing the outcomes of this project, the evaluation will contribute valuable insights into the role of community engagement in enhancing public health initiatives and overcoming persistent health challenges exacerbated by global crises such as the COVID-19 pandemic.

The Project adapts a novel and collaborative approach by the complementary use of CBPR and HCD approaches in these two contexts to identify drivers and co-create context-specific and community-driven solutions with intended users (caregivers and healthcare workers) in

order to generate demand and address obstacles in immunization access and uptake. Therefore, this evaluation is key and will contribute to this field by evaluating the effectiveness of the project on immunization outcomes and its implementation through mixed methods research. It will potentially provide evidence on mechanisms that explain observed improvements in under two immunization outcomes and toward resilient health systems as a result of the co-created community-driven solution(s).

## Materials and methods

### Aim of the evaluation

The aim of this study is to evaluate the effectiveness of the Project on under two immunization coverage and to assess the extent of its implementation (reach, adoption, implementation and maintenance) in Mozambique and Malawi.

A hybrid evaluation focuses on both effectiveness and implementation [36]. We chose to use a hybrid evaluation embedded within the project. This approach provides an opportunity to work together with the implementing partners while simultaneously evaluating both the effectiveness of the intervention on specific outcomes of interest (impact) and the extent to which the intervention reaches specific implementation outcomes such as reach, fidelity, acceptability, sustainability [36, 37].

### Objectives

We specifically aim to provide evidence over a three-year period guided by the Reach, Effectiveness, Adoption, Implementation, and Maintenance (RE-AIM) framework as follows:

### Evaluation of the effectiveness of the co-created interventions:

**Objective 1:** To assess the impact of the co-created interventions on routine under two immunization outcomes at baseline, midline and end line of the pilot implementation.

### Evaluation of the implementation of the Project:

**Objective 2:** To conduct a process evaluation to assess the implementation of the participatory solution development and solution implementation.

### Development of a comprehensive framework for community responsive programming:

**Objective 3:** Gather evidence to inform and develop a comprehensive framework on community-responsive programming in routine under two immunization to guide the scalability of community-based approaches for improving childhood immunization uptake and access into similar LMIC contexts.

### Study design

Guided by the RE-AIM framework, a prospective quasi-experimental and process evaluation of outcomes and processes will be conducted using mixed methods to prospectively evaluate the effectiveness (on under-two immunization coverage) of the intervention (participatory solution development and solution implementation) and processes of implementation (reach, adoption, implementation and maintenance). The RE-AIM framework is used in planning and evaluation, operating according to five dimensions; Reach, Effectiveness, Adoption, Implementation and Maintenance [38, 39]. Each aspect of the RE-AIM framework will be

systematically evaluated through our mixed-methods approach including document and literature review, in-depth interviews, observations, surveys and analysis of quantitative data sets. RE-AIM as an evaluation framework helps to provide an overall descriptive account of the intervention elements by defining implementation outcomes for sustainable and effective implementation [38, 39].

The study will also draw on selected implementation science constructs from the consolidated framework for implementation research (CFIR) to understand the factors influencing implementation. The CFIR framework is an explanatory framework outlining effective implementation consisting of 39 constructs organized into five domains: Intervention Characteristic; Outer Setting; Inner Setting; Characteristics of Individuals; and Process [40, 41]. A theory of change (TOC) will articulate the "path" to outcome generation for the identified implementation strategies of the Project, summarized in Fig 1.

The sequence of the evaluation is summarized in Fig 2.

## Study setting

The study will be conducted at the Project's intervention sites (health facilities and catchment areas) in four selected districts, two in Malawi and two in Mozambique located in Southern Africa. Intervention districts were selected by VillageReach to participate in the participatory solution development and implementation of the co-created interventions. In Mozambique, the evaluation will be conducted in Namarroi and Gile (both rural districts) located in Zambezia Province. In Malawi, the study will be conducted in 1 rural setting in Mzimba North and 1 urban informal setting in Lilongwe. Control sites for comparison will be selected in coordination with VillageReach staff and national, provincial and district Expanded Immunization Programme (EPI) leadership. Control sites will be selected once the co-created interventions have been developed following the CBPR and HCD process. To select control sites, we will consider variables such as comparable routine immunization rates, average walking distance to health

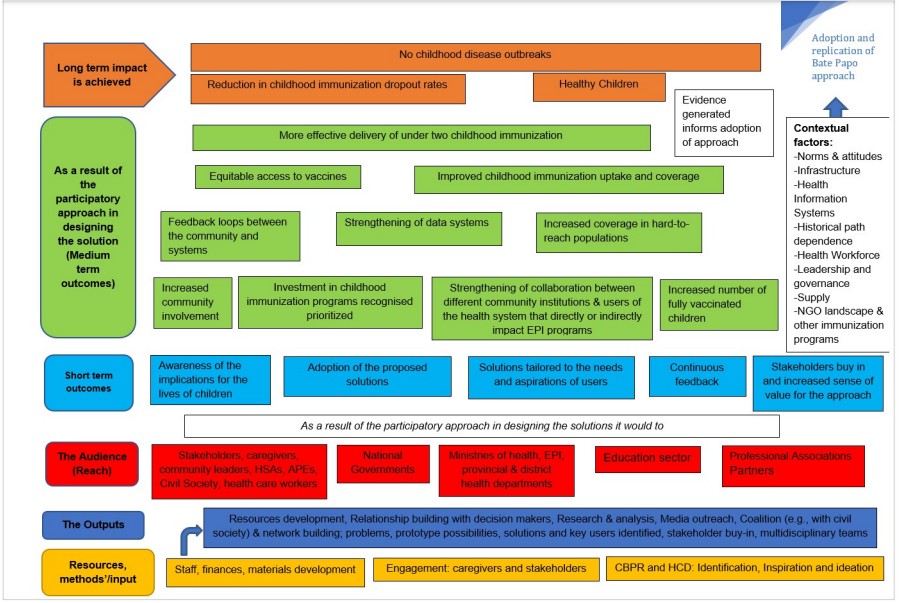

**Fig 1. Theory of change.**

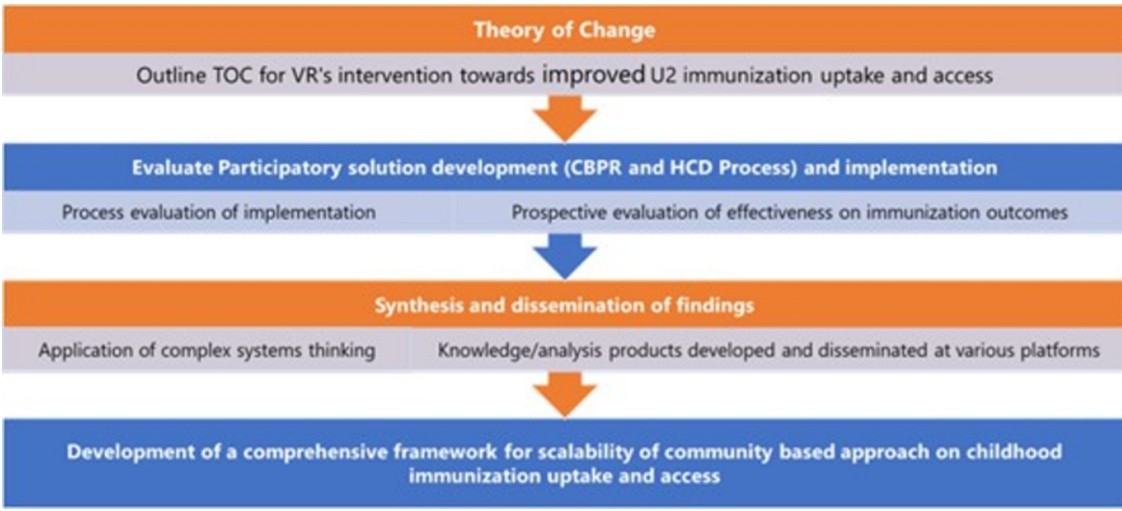

**Fig 2. Sequence of evaluation.**

facilities, geography, population size, other Non-Governmental Organizations (NGOs) operating and other research studies and/or similar programs.

## Study population

*For the evaluation of the effectiveness of the co-created interventions*, we will use de-identified aggregate data on under two routine vaccination outcomes. This population includes all children under the age of two eligible for vaccination within the intervention and control sites. Vaccines being studied are all the under-two vaccines as stipulated in the vaccine schedules of the two countries [42, 43].

*For the evaluation of the implementation*, the study population includes participants involved in the CBPR and HCD approach as well as the implementation of the co-created interventions in the four selected districts based on VillageReach's inclusion and exclusion criteria of the participants (see Table 1). This includes caregivers, caregiver researchers, health care workers (HCWs), EPI officials at the district, provincial and national levels, community leaders, the VillageReach project, national and global staff and other relevant stakeholders involved in immunization programming, or who can influence immunization access and uptake.

## Study timeline

The evaluation spans from October 2022 onwards and will continue until December 2024. There is a possibility of extension until March 2025 to accommodate ethics extension applications and approvals at the country level, any changes in timelines for each country, unforeseen contextual factors and the completion of framework development.

## Sampling

*Evaluation of the effectiveness of the co-created interventions*, all de-identified aggregate records on under two childhood immunization, from 2022 to 2024 will be sampled for the quantitative data (impact indicators). To compare outcomes of interest, the records will be considered from selected intervention and control sites in the districts at baseline, midline, and end line of the pilot implementation of the co-created interventions. For comparison, the control sites

**Table 1. Inclusion and exclusion criteria for the participatory solution development.**

*Inclusion criteria*

• Caregiver researchers: these are women from the intervention districts who are caregivers of young children, who were hired and trained by VillageReach to conduct the participatory data collection activities, participate in data analysis and solution design. Standardization of the role of caregiver researchers across all sites will include consistent training for the necessary skills, knowledge, and understanding of research protocols; consistent data collection tools; defining and maintaining consistent roles and responsibilities throughout the study and quality control of data collected through regular feedback process throughout the study period.

• Participants who were part of the CBPR study and HCD and/or implementation of the co-created interventions as follows:

  ○ Health care workers specifically responsible for vaccination education and administration.

  ○ Caregivers of young children. Caregivers include parents or other family members or guardians who take the primary responsibility of bringing the child to access health services. Caregivers must have a child aged 25–34 months with health center documentation of at least one of the routine under two vaccinations completed by age 24 months as well as contact information (phone number, house directions). An eligible participant must be the primary caregiver of the child in question, meaning that they are the guardian who takes the primary responsibility of bringing the child to access health services.

  ○ National, provincial and district health officials from the selected districts in Mozambique and Malawi (Gile and Namarroi; Mzimba North and Lilongwe).

  ○ Community leaders who participated in the Project from the intervention sites.

• VillageReach project staff who were historically involved or currently involved in the study and implementation of the co-created interventions.

*Exclusion criteria*

• Caregivers who were not employed by VillageReach as a Caregiver Researcher.

• Health care workers who do not provide vaccines and have no experience of vaccinating children.

• Caregivers with no documentation of any vaccinations for their child; if their child is under the age of 25 months or over the age of 34 months and caregivers who are under the age of 18 years.

• National, provincial and district health officials who have no responsibilities related to routine immunization and have not participated in the Project.

• Community leaders who were not involved in the Project and if they are not from the intervention sites.

• VillageReach staff who have not been involved in the conceptualization, design or implementation of the project.

should be as similar as possible to the intervention sites in terms of variables other than exposure to the intervention. For this project, the control sites need to be matched to the intervention sites in terms of the under-two immunization rates, distance to the health facilities, catchment population and if information is available, number of health care providers and other NGOs operating in the districts.

*For evaluating implementation*, purposive sampling will be employed in consultation with VillageReach to select caregiver researchers, caregivers, HCWs, VillageReach project staff, community leaders and other key stakeholders involved in the Project. Purposive sampling will target participants involved in the CBPR and HCD approaches, as well as those involved in implementing the piloted co-created interventions at midline and endline points in both countries. Participants will be selected based on the criteria in Table 1. In addition to participant selection, researchers will gather relevant documents from VillageReach, search databases and use a snowballing approach to identify articles and other relevant documents [44].

## Sample size

The following sample size will be considered:

**Evaluating the effectiveness of the co-created interventions.** To be able to conduct an impact model (as described in the data analysis), a hypothesis about the dropout rates will be proposed at the start, depicting the possible effectiveness of the intervention. We thus present here a rough sample size calculation subject to revisions once a decision has been made on "proxy" vaccines to be used to develop the matrix for the 'dropout rates' outcome (specific

under two childhood vaccines will be considered for monitoring the dropout rates as some vaccines have full coverage in the different study sites).

Our quantitative sample size will constitute all available data on under two immunization, which will be bounded by time (per month) and geographical location as specified for the intervention and control sites. Based on a recent study in Malawi [43], the average dropout rates for 2015–2016 using diphtheria, tetanus, and pertussis 1 (DPT1) - pentavalent 1 (Penta1) and DPT3-Penta3 and DPT1-Penta1 and measles-containing vaccine 1 (MCV1)-measles and rubella 1 (MR1) were 4.5 and 6.4, respectively, giving an average of 5.5% ~6% [43]. For a 1-sided test with a 5% significance level, power of 80% and a 50% change in effect size (dropout rate reduction from 6 to 3%), a sample size of 286 is needed. A 50% reduction in dropout rates is a conservative estimate to avoid a type II error. A total sample size of 300 observations will be used for the impact model analysis to allow for any data quality issues and other threats to obtaining a well-powered sample size. The same calculation will be hypothesized for study sites in Mozambique, as accurate statistics on the actual dropout rates for the same vaccines used here could not be established.

**Evaluating the implementation.**   We plan to recruit a total of 120 participants for the interviews inclusive of both countries. These include 10 caregivers, 10 HCWs and 10 key informants (KIs) from national, provincial, district and local/community stakeholders as well as VillageReach project staff, in each of the four intervention districts. To recruit participants for interviews, we will contact potential participants via phone, email or personal contact where possible and explain the purpose of the study and interview before asking if they would be willing to participate. Participation in the study will be voluntary, and all participants who agree to participate will be asked to sign written consent. Data saturation will be considered when an adequate sample size is reached [45]. Table 2 below shows a breakdown of the sample size according to the selected districts.

## Data collection

Data will be collected concurrently to evaluate the effectiveness on under two immunization outcomes and the extent of the Project in reaching specific process and implementation outcomes, as follows:

**Documents and literature review.**   As an ongoing process, we will conduct an extensive review of the literature and documents on CBPR and HCD practice, under two immunization coverage and health systems resilience in line with the evaluation. Existing documents in

**Table 2. Sample size according to category and districts/location.**

|  | Health Care Workers | Caregivers | Community leaders | District officials | Sum |
|---|---|---|---|---|---|
| Mzimba district | 5 | 10 | 5 | 3 | 23 |
| Lilongwe district | 5 | 10 | 5 | 3 | 23 |
| Provincial health staff |  |  |  |  | 4 |
| VillageReach Staff |  |  |  |  | 3 |
| Caregiver Researchers |  |  |  |  | 4 |
| Gile district | 5 | 10 | 5 | 3 | 23 |
| Namarroi district | 5 | 10 | 5 | 3 | 23 |
| Provincial health staff |  |  |  |  | 4 |
| VillageReach Staff |  |  |  |  | 3 |
| Caregiver Researchers |  |  |  |  | 4 |
| VillageReach staff |  |  |  |  | 6 |
| **Total** |  |  |  |  | **120** |

Malawi and Mozambique, including grey literature from VillageReach, such as project reports that include attendance records, CBPR results, meeting notes from prototyping activities, emails, presentations, records from health facilities and Demographic and Health Survey (DHS) reports, will be sourced from VillageReach staff, database searches and snowballing of articles. Documents will be stored and organized in Mendeley, and a matrix will be developed to summarize information gleaned from the reviewed documents [46].

**Semi-structured interviews.**   We will conduct interviews using a semi-structured guide with 10 caregivers, 10 HCWs workers, and 10 KIs (country-based and global) in each setting, involved in the Project. Interviews and knowledge, attitude and practice surveys (KAPs) with caregivers, HCWs, project staff, and other key stakeholders will be conducted as ongoing research on implementation outcomes across contexts at the midline and end line of the implementation. The study will be explained to the participants, and they will be provided with an information sheet. Informed consent will be obtained from participants by each signing a consent form.

**Non-participant observations.**   An observation guide will be used during the HCD workshops to capture the depth and fast pace of the process (engagement, identification of drivers and solutions, etc.) [16]. The purpose of the observations is to observe the engagement with all the participants involved in these workshops. As such, no names will be put down, but we will merely note the level of participation by categories of stakeholders.

**HCD participant feedback survey.**   We will also obtain immediate feedback at the end of each HCD workshop using a feedback survey form. The survey form was adapted from [18] and adjusted to suit the objectives of this evaluation [17].

**Collection of health and demographic data on under two immunization.**   The evaluation team will obtain health and demographic data on under two immunization dropout rates and other outcome measures from various sources at different time points (pre-intervention, mid-way, and end of intervention) in both countries in the intervention and control sites. Under two is the standard timeframe looked at for assessing routine immunization coverage and key indicators as most children should receive all required vaccination by the age of 2 [1]. This is also aligned with the recommended vaccination schedule for both Mozambique and Malawi [42, 43]. Data sources will include administrative data extracted from the Malawi Demographic and Health Survey (MDHS) and the Health Information System of Mozambique for Monitoring and Evaluation (SIS-MA) through district reports, health facility records, online and any other available data sources.

Immunization data will be aggregated from multiple health centers. Data collection, entry and processing typically face quality and completeness challenges due to a lack of data management infrastructure and a lack of well-trained staff. However, we assume that these challenges will occur at a random manner in control and intervention sites as the intervention does not target improvement of data management. As such, we assume that these challenges will not affect the study arms disproportionality, but may result in a lack of precision. We will conduct a data quality assessment and develop an appropriate mitigation plan prior to beginning the full evaluation. This plan will include the implementation of data validation techniques such as cross-checking between multiple data sources such as electronic databases, health facility data and monitoring data on the relevant outcomes, considering the potential data challenges in the given contexts. Data collected by the project will be used as a quality standard to compare and evaluate potential data quality problems when collected through the health system. Furthermore, site visits will be done to evaluate the data collection and processing pathway with a focus on procedural compliance and consistency across sites. Finally, data received by our team will be checked for inconsistencies, outliers, missingness and other traceable errors.

Interrupted time series analysis integrates multiple datapoints which will enable to address problems of data integrity.

The above would entail working closely with the VillageReach staff, and EPI officials within the selected districts to support data verification and ensure documentation of any anomalies and limitations. In addition, to address potential data quality challenges with administrative data, the primary outcome of interest will be dropout rates, which does not require having to rely on (often contested) population data. Dropout rates give an indication of the success of vaccination throughout the required stages. The WHO defines dropout rates as "the percentage of children that started their immunization series but didn't finish it for some reason" [47]. It is calculated as the "[(coverage of initial vaccine dose minus coverage of ending vaccine) divided by (coverage of initial dose) x 100]" [48]. Proxy vaccines will be agreed upon for measuring the outcomes depending on the under two immunization schedules for Mozambique and Malawi.

## Data analysis

Document analysis will identify, sort and organize data from the documents related to HCD practice, CBPR, under two immunization and the Project into specific themes [46].

Interviews will be tape-recorded and transcribed verbatim for analysis. Atlas.ti will be used to capture the qualitative data and thematic analysis applied to analyze the data inductively and deductively with the RE-AIM domains, CBPR/HCD principles, relevant CFIR constructs and reference to the TOC. The data will be coded and categories and themes developed to make sense of the data. Data from observations and surveys will be used to add to the thick description of the analysis. Complex systems thinking will be employed to assess the embeddedness of the intervention within a wider immunization ecosystem, as well as relationships and mechanisms that explain emerging implementation and impact/effectiveness-related outcomes (intended and unintended consequences). Health care is seen as a complex adaptive system (CAS) that has interactions and interconnectedness among its different parts (professionals, patients, equipment) leading to health care delivery. This complexity can influence the impact of interventions as a reaction to internal and external catalysts [49]. Additionally, a CAS lens will guide the interpretation of the analysis with the use of constructs embedded in CAS, such as phase transitions, feedback loops, historical path dependence, self-organization, existing norms and attitudes, in response to the identified interventions [50]. Analyzed qualitative data will be used to form a holistic understanding of effectiveness and inform the development of appropriate quantitative indicators. Qualitative data will include caregivers' acceptance of the interventions as well as insights into the implementation processes, adaptability and sustainability of solutions.

An interrupted time series analysis will be used to analyze the under two immunization dropout rates and other outcomes of interest. A time series analysis is a quasi-experimental design that involves the collection of data by creating a time series of an outcome at equally sized time points and statistically testing for changes in the outcome pre- and post-intervention [51, 52]. The exact time of interception of the intervention should be known. In this study, information will be obtained from VillageReach when the pilot implementation of the co-created interventions will commence in both countries. Under two immunization outcomes will be analyzed in the study settings in Malawi and Mozambique from the period prior to implementation (baseline), midway of the intervention and at the end line of intervention. Data collected from the control sites will be analyzed to depict predicted trends (counterfactual scenarios of change without the intervention) for comparison with data where the intervention took place [53]. This will reduce bias from any time-varying confounders.

Overall, with the interrupted time series analysis, an impact model will be proposed as a hypothesis at the start, depicting possible effectiveness of the intervention on the outcome (e.g., immediate level change or a lag period before effect) based on the data [53]. For seasonal adjustment, the model will be stratified according to calendar months. Segmented regression will be used to statistically analyze the level (immediate changes in the dropout rates) and slope (changes in trend) compared pre-intervention and post-intervention [52]. The results will be presented in the form of scatter plots and line graphs to show the trends.

The above data collection process and the data analysis will assist in documenting communities' perceptions and experiences of their involvement in participatory solution development and whether the interventions identified align with their needs and opinions. A CAS lens will also help uncover important drivers of change for a reduction in immunization dropout rates and immunization uptake and how innovative context-based solutions developed include health system strengthening activities for sustainability and health system resilience. Illustrative feedback loop diagrams will be used to show the interactions and connections among activities and partners that result in changes in immunization coverage [50, 54]. This analysis will set a basis for developing a comprehensive framework to guide the application of the community-based approach to childhood immunization uptake in similar LMIC contexts.

## Trustworthiness and credibility

For credibility, respondent validation will be used to acquire reflections from participants on preliminary findings, interpretations, and consistency with their experiences. Triangulation of data sources (interviews, participant observations, surveys, etc.) will also be applied for a detailed set of results and a thick description of the findings [55].

## Ethics approval and consent to participate

This evaluation received ethics approval from the Biomedical Science Research Ethics Committee of the University of the Western Cape (Ethics Reference Number: BM22/4/3), the Mozambique National Bioethics Committee for Health (Reference: 588/CNBS/22) and the National Health Sciences Research Committee of Malawi (Protocol # 22/08/2987). The study will be conducted in accordance with the general ethical guidelines and regulations of the ethics committees. Informed consent and anonymity/confidentiality will be adhered to during the study. There are no anticipated negative consequences from participation in the evaluation; thus, there is minimal risk. We will regularly review and revise ethical considerations during the study, especially regarding participant privacy and data sharing agreements in accordance with ethics requirements in both countries and evolving changes in public health research.

## Outcomes- effectiveness, and implementation

Outcomes will be assessed at baseline before the pilot of the co-created interventions, at midline and end line. The outcomes collectively represent various elements of analysis for effectiveness and implementation (see Tables 3 and 4). Studying outcomes in each context enables comparison and cross-contextual learning. Outcome indicators across different contexts within countries (intervention and non-intervention) and across country sites will be compared.

**Table 3. Under two immunization outcomes of interest for effectiveness evaluation.**

| Immunization outcome | Outcome description and interpretation |
|---|---|
| * Diphtheria, tetanus, and pertussis (DTP)1-DTP3 under two immunization dropout rates | DTP1–DTP3 immunization dropout rates at the selected intervention and control health facilities. Dropout rate is calculated as follows: (# of children who received initial vaccine dose minus # of children who received 3rd dose) divided by (# of children who received initial dose) x 100]. A decrease in dropout rate implies improved utilization and increase implies poor utilization. |
| Number of zero dose children | # of children who did not receive any routine vaccine/ Children who have not received a single dose of DTP. Decrease in zero dose children implies reach and improved coverage. |
| % under two immunization coverage rates by antigen | # of children immunized during the last 12 months by antigen divided by number of children eligible (x 100). Increased % implies availability, access to and continuity of use. |
| % stock out rates per health facility | Product/antigen absence over a given period (# of facilities that experienced a stockout of a specific vaccine divided total # of facilities expected to provide vaccine x 100). High stock out rates implies problems in the supply chain and disruption of services. |
| Number of children vaccinated at the selected intervention and control health facilities and from the outreach sessions or mobile brigades | Absolute numbers of children vaccinated at the selected intervention and control health facilities and from the outreach sessions/mobile brigades |
| % of fully vaccinated children | Full coverage indicator (not for a specific antigen) to designate a child who completes the vaccination calendar during the 11 months of life. |
| Number of mobile brigades/outreach sessions planned and implemented; | Indicator depicting number of actual mobile brigades executed (difference between number of mobile brigades/ outreach sessions planned versus executed) |
| *Pentavalent 3 (Penta 3)-*Measles-containing vaccine 1 (MCV1) dropout rate | Measles vaccination dropout. Penta1 to MCV1 should be <10%. |
| Dropout rate for first to second dose of measles containing vaccine | MCV1 to MCV2 dropout = 100x (MCV1-MCV2)/MCV1. The MCV1 to MCV2 dropout rate assesses the ability of the program to vaccinate beyond the first year of life |

*DTP, diphtheria, tetanus, and pertussis; MCV, Measles-containing vaccines; Penta, Pentavalent.

## Theory of change

The theory of change maps the "path" to outcome generation of the Project and will be refined as more evidence is generated during the evaluation. As a result of the participatory solution development, short-term outcomes include awareness of the implications for the lives of children under two; adoption of the proposed solutions; solutions tailored to the needs and aspirations of both the individuals who use the public immunization system (caregivers of children under two) and those who deliver the care (health care workers and Agentes Polivalentes Elementares (APEs) in Mozambique or Health Surveillance Assistants in Malawi); continuous feedback; stakeholders' buy-in and an increased sense of value for the CBPR and HCD approach.

Furthermore, medium-term outcomes foreseen as solution implementation commences include increased community involvement, investment in childhood immunization programs and recognized priorities, improvement in the number of fully vaccinated children and coverage in hard-to-reach populations. Closely interrelated is the strengthening of collaboration between different community institutions and users of the health system that directly or

**Table 4. Implementation outcomes for the evaluation of implementation.**

| RE-AIM Domain | Evaluation Focus |
|---|---|
| Reach: The absolute number, proportion, and representativeness of individuals who are willing to participate in a given initiative, intervention, or program [30]. | It is important to have a defined target population to ensure that the implementation strategies improve access to and uptake of under two immunizations. |
| | Furthermore, the intervention should have elements that are appropriate to meet the target audience's needs. *Appropriateness*–the perceived relevance and fit of the solutions, from the perspective of caregivers and service providers. This is particularly important for interventions that for, instance, may have service implications, such as changing how service providers give immunization services. |
| Effectiveness: understanding the comprehensive effects of a program, including unintended consequences | In this study, the effect of the intervention on immunization coverage and other measures will be assessed (see outcomes of interest in Table 3). |
| Adoption: the measurement of the uptake of interventions in each context, from the perspective of the intended target populations, such as caregivers and health service providers. | To be adopted, the intervention has to be acceptable, i.e., whether the approach for community engagement was considered agreeable/satisfactory. Acceptability will be explored from the perspectives of the intended beneficiaries and stakeholders, particularly the caregivers, health service providers and others involved in the *HCD process. |
| | Furthermore, adoption also speaks to the feasibility of the suggested intervention in a given context. Strictly defined, feasibility is a retrospective look at the organizational requirements for implementing the solution successfully within a defined setting, in order to inform implementation strategy in other contexts. |
| Implementation: focuses on fidelity to an intervention: the extent to which the program is implemented consistently across different settings, staff, and patients. | Fidelity in this study refers to the use and adherence to protocol, where the CBPR and HCD process and the interventions proposed, are implemented as designed including any adaptations made. In other words, how well VillageReach adhered to the *CBPR and HCD process and the interventions as designed. |
| Maintenance: has indices at the individual- (long-term effectiveness - and extent to which behavior is sustained 6 months or more after the intervention) and setting-level (sustainability beyond external funding). Maintenance also considers the "diffusion" of an innovation, whether the intervention(s) has been integrated at the service/system level and how daily practice is influenced. | Sustainability (maintenance) requires a longer time frame and may not be realized in the given project time frame. |
| | However, there are some aspects of sustainability that might be observed as the implementing partner aims to involve local ministries of health when creating interventions. There is thus a reasonable possibility of Let's talk about vaccines! approach and solution adoption and integration at the system/service level. |

*HCD, Human centered design; CBPR, Community Based Participatory Research.

indirectly impact EPI programs, which facilitates feedback loops between communities and systems such as health care, education, NGOs and the strengthening of data systems for improved reporting of childhood immunization coverage. In an intertwined process, there will be equitable access to vaccines and improvement in under two immunization uptake and access in specific settings. This results in the effective delivery of under two childhood immunization which in turn leads to a reduction in dropout rates, an indicator of the success of immunization interventions [56]. When children are fully vaccinated, long-term impacts, such as a state of healthy children and the absence of childhood disease outbreaks, will be observed.

## Context

Adoption and replication of the Project will also consider the contextual factors prevalent within the healthcare system, such as norms and attitudes and historical path dependence. Phase transitions, self-organization and emergent behavior and health systems issues such as infrastructure, health information systems, financing, health workforce, leadership and governance, service delivery and supply chain issues will play a role in the adoption of the Project. Other contextual factors include the following:

- Inequalities in childhood vaccination coverage within and among countries despite advances in childhood immunization programmes globally such as EPI.

- Childhood immunization uptake and access is low, especially in hard-to-reach populations.

- New solutions driven by international stakeholders with limited caregiver and health worker engagement in identifying barriers and solutions to improve vaccination coverage.

## Assumptions

To improve childhood immunization uptake and access and reduce childhood disease outbreaks, we assume the following:

- A participatory co-creation process in identifying barriers to childhood immunization and solutions will reduce vaccination dropouts, and by using this tailored participatory approach to solution identification and design, the solution would have high adoption by users.

- Tailored solutions that meet the needs and aspirations of caregivers will improve childhood immunization uptake and access.

## Dissemination

The results of the evaluation will be shared regularly with VillageReach and relevant stakeholders in the Project. The findings will be disseminated to local, provincial and national governments, implementers, funders and relevant organizations to communicate lessons learned from the process of identifying barriers and potential solutions to under-two immunization. Knowledge/analysis products will be developed in the form of reports, policy and technical briefs, and scientific articles and distributed through scientific and media platforms (such as the MESH community engagement network and Boost community), community engagement, stakeholder forums, academic conferences and local and national governments.

# Discussion

## Significance

This evaluation is an opportunity to discuss the use of novel and best practices, opportunities and challenges for improving community-responsive programming in routine immunization and scalability to similar contexts. Guided by the RE-AIM framework we aim to collect information for evidence translation by clearly defining implementation outcomes, explaining factors that influence the outcomes (CFIR), and outlining evidence of the effectiveness of the Project on under two immunization indicators that can be compared across different contexts and provide evidence to improve implementation and inform the scale-up of activities.

There are several opportunities to build and refine the theory of change. At the beginning where solutions are developed in collaboration with the community, there is useful evidence for how stakeholders theorized on desired outcomes and how the proposed solutions are

expected to produce them. As the implementation of solutions is ongoing, the evaluation will provide more nuanced insight into contextual factors, actor interactions, and other factors that will ultimately influence effectiveness. In the end, recommendations on the adaptation of the Project to other contexts will include evidence on its effectiveness and how it was influenced by context and implementation factors.

## Limitations

This is a multi-site evaluation and is prone to some limitations, which include the following:

There is a possibility of incomplete records from already collected routine immunization data and data from other sources. This may affect the interrupted time series analysis for comparisons pre-intervention, midway, and end-line of the intervention. A data quality assessment plan will be developed, and the evaluation team will collaborate with VillageReach staff, EPI officials at district and health facility level to validate data quality. We will also document any anomalies to aid future research.

Since the evaluation has a multi-site component, precise alignment with the iterative fast-paced process of the approach may be affected by situational and contextual factors in each district.

The evaluation may result in some form of bias, as participants' responses may be slightly altered as a reaction to an evaluation of processes that they are part of and responsible for.

## Supporting information

**S1 Table. Proposed checklist.**
(PDF)

## Acknowledgments

The authors acknowledge VillageReach staff at the global and country levels for their support in providing any valuable information regarding this study protocol.

The authors would like to acknowledge funding from Wellcome Trust.

LS and HS are partially supported through the South African Research Chairs Initiative of the Department of Science and Technology and National Research Foundation of South Africa (grant no. 98918).

## Author Contributions

**Conceptualization:** Linda Shuro, Emily Lawrence, Lucia Knight, Helen Schneider, Hanani Tabana.

**Funding acquisition:** Emily Lawrence.

**Methodology:** Linda Shuro, Lucia Knight, Helen Schneider, Hanani Tabana.

**Writing – original draft:** Linda Shuro.

**Writing – review & editing:** Linda Shuro, Emily Lawrence, Lucia Knight, Helen Schneider, Hanani Tabana.

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
