## [Decision Letter · Decision Letter 0]

21 Jun 2024

PONE-D-24-15369Enhancing Childhood Immunization Coverage in Mozambique and Malawi: Study Protocol of a mixed methods evaluation of the Let's Talk About Vaccines Multisite Community-Based Participatory ProjectPLOS ONE

Dear Dr. Shuro,

Thank you for submitting your manuscript to PLOS ONE. After careful consideration, we feel that it has merit but does not fully meet PLOS ONE’s publication criteria as it currently stands. Therefore, we invite you to submit a revised version of the manuscript that addresses the points raised during the review process.

We look forward to receiving your revised manuscript.

Kind regards,

Edison Arwanire Mworozi, M.D

Academic Editor

PLOS ONE

Journal Requirements:

Additional Editor Comments:

Please revise this document as per reviewers comments!

Reviewers' comments:

Reviewer's Responses to Questions

**Comments to the Author**

1. Does the manuscript provide a valid rationale for the proposed study, with clearly identified and justified research questions?

Reviewer #1: Partly

Reviewer #2: Yes

Reviewer #3: Yes

Reviewer #4: Yes

Reviewer #5: Yes

Reviewer #6: Partly

Reviewer #7: Partly

2. Is the protocol technically sound and planned in a manner that will lead to a meaningful outcome and allow testing the stated hypotheses?

Reviewer #1: Partly

Reviewer #2: Yes

Reviewer #3: Yes

Reviewer #4: Yes

Reviewer #5: Partly

Reviewer #6: Partly

Reviewer #7: Partly

3. Is the methodology feasible and described in sufficient detail to allow the work to be replicable?

Reviewer #1: Yes

Reviewer #2: Yes

Reviewer #3: Yes

Reviewer #4: Yes

Reviewer #5: No

Reviewer #6: Yes

Reviewer #7: No

4. Have the authors described where all data underlying the findings will be made available when the study is complete?

Reviewer #1: Yes

Reviewer #2: Yes

Reviewer #3: Yes

Reviewer #4: Yes

Reviewer #5: Yes

Reviewer #6: Yes

Reviewer #7: No

5. Is the manuscript presented in an intelligible fashion and written in standard English?

Reviewer #1: Yes

Reviewer #2: Yes

Reviewer #3: Yes

Reviewer #4: Yes

Reviewer #5: Yes

Reviewer #6: Yes

Reviewer #7: Yes

6. Review Comments to the Author

You may also provide optional suggestions and comments to authors that they might find helpful in planning their study.

Reviewer #1: General comments

Vaccinations remain a major advancement in global public health, despite challenges for these live-saving vaccines to reach a majority of children at need in hard-to-reach zones. It is important to understand community-level drivers on vaccine uptake needed to support health decision making about prioritization and cost of these interventions in one’s own setting.

This study protocol aims to evaluate the effectiveness (on under-two immunization coverage) and the extent of implementation (reach, adoption, implementation and maintenance) of the Project in Mozambique and Malawi.

Authors plan to integrate the theory of change approach with the RE-AIM framework, and hope findings will potentially provide evidence on mechanisms that explain observed improvements in under two immunization outcomes and toward resilient health systems as a result of the co-created community driven solution(s).

The paper will bring important addition to existing literature, especially from an area where data paucity remains a perennial issue. However, this reviewer finds that there are gaps to address in the current version of the paper before it is considered for publication.

SPECIFIC COMMENTS

Title: No need to capitalize each word, only names of places and persons here except that falls within the journal’s specifications.

- The manuscript lacks line numbering for easy reading and review. Also, there is need for authors to format the entire work appropriately.

ABSTRACT

- CBPR: Please after first use of an abbreviation, it should be used thereafter. No need spelling out in full.

- “There is a few example of evaluations of intervention created through human-centred design.” Authors can integrate this in the next sentence i.e., “… this approach...” or take it to the introduction.

INTRODUCTION

- “It is an approach that centres on creativity in an interactive process to bring human centred views … “Here, I think it gives a wrong impression that previous interventions have not been human-centred. This should be revised to community-directed/driven/centred.

- “… by evaluating the effectiveness and implementation of the project.” How will these be performed?

- Ensure consistency in reporting (e.g., RE-AIM or RE AIM)

- In figures/ Tables, authors should ensure that each abbreviations should be fully defined as footnotes if not within the figure (s)/Table(s) e.g., what is TOC, VR, HCD?

- DPT1-Penta and all abbreviations should be fully defined at first use before subsequent uses.

- Could the authors describe how data quality will be assessed considering the porous nature of data in resource-low settings?

- No need to put “long-term impacts” in bold!

- Figure 2 is very illustrative and it’s great! Except that some of the arrows are out of place or non-directional. Please, verify and adjust!

- What’s the rationale of targeting under-2 populations? Would this be sufficient to assess the long-term impacts?

- Ethics statement should only appear in the text (not abstract, except the journal demands that).

DISCUSSION/Limitation

- Incomplete records are a perennial problem in resource-low setting, how do the authors plan to ascertain data quality with this huge challenge?

- Additionally, authors need to explain clearly how they plan to overcome the stated limitation so as not to jeopardize potential findings.

REFERENCES

- Some are wrongly captured and should be verified at the level of the reference manager before uploading, so that articles should be referenced appropriately.

Reviewer #2: The standardization of the caretaker researchers and the control sites is missing. There is also need to start the criteria for the purposive sampling at the midline assessment. Study data quality control and assurance measures are wanting.

Reviewer #3: The manuscript outlines a detailed and systematic protocol for assessing the effectiveness and implementation of the "Let’s Talk About Vaccines" project. Drawing from my extensive experience in global health security and epidemic intelligence, I find the use of CBPR and HCD methods both innovative and highly appropriate for addressing the critical issue of low childhood immunization rates in Mozambique and Malawi. The methodology is robust and well-documented, ensuring that the study can be replicated and will yield valuable insights.

Here are a few minor recommendations for improvement:

1. Continuously review and update ethical considerations throughout the study, particularly concerning participant privacy and data sharing agreements, reflecting the dynamic nature of public health research.

2. Provide more specific examples of successful applications of the HCD approach in similar contexts to better justify its use in this study. My work with various health security initiatives has shown the importance of such context-specific illustrations.

3. Address potential limitations related to the interrupted time series analysis, especially concerning the quality and completeness of routine immunization data. Drawing from my experience in managing large-scale health data projects, ensuring data integrity is crucial for accurate outcomes.

Overall, the manuscript is comprehensive and well-prepared, setting a solid foundation for subsequent stages of the review process.

Reviewer #4: Background:Add more context:

1. Definition of vaccine coverage.

2. Highlight the challenges of vaccine coverage.

3. Mention the challenges that the methods under study are aimed to address.

Methods: Study population:

2. Which vaccines are being studied? Are the challenges of vaccine coverage the same for all vaccines?

Results/Dissemination:

Is health education/community dissemination in the plan?

Reviewer #5: Overall, relevant study area. The objectives can be explained more explicitly. The methodology, especially the sampling to be more detailed.

Reviewer #6: The author must explain rationale for the study more clearly, just referring to sub-optimal full immunization coverage and COVID-19 effect does not serve the purpose. It should define if there are some barriers affecting reaching to community, any particular beliefs etc. that hinder vaccination coverage.

Also what proposed intervention the Village Reach team expect, out of the HCD workshops planned? More clarity on possible intervention that may come out must be mentioned here. If this study has been conducted in other countries of LMIC/LIC, it would be useful to quote some examples.

Reviewer #7: Due to the process-oriented nature of the proposed analysis, the protocol as described in the current manuscript is not replicable. Given that replicability is the manuscript's primary objective, its contribution to the existing body of literature remains ambiguous.

A revision would be stronger if the methods were more specific and much more concise. The methodology may not lend itself to such a description. Remove didactic language. While many theories of evaluation are referenced it is unclear how they are all brought together into a cohesive analytic plan.

It is unlikely that the study is powered appropriately to detect significance of impact on a dropout rate of 5–6%. A sample size calculation to see a difference of about 5% coverage usually needs around N=250. You would be looking for a change of 2%. The study would not need a 2-sided test unless there is a reason why the proposed interventions would increase dropout rates? Is the "adequate sample size" for the study as a whole, for each country or each site?

The document does not describe the full timeline of the proposed study the timeline section states "until the end of 2024" when did the study start? How will the midpoint be determined?

Given that the current only baseline and two assessment points, how can you stratify the model by month?

Refine the terminology around assumptions: in the introduction and in the assumptions section you are putting forward hypotheses. Assumptions are caveats and determinations that are not part of the formal study outcomes.

Theory of change diagram: while rightly recognizing the complexity of healthcare utilization and delivery, the diagram and theory should reflect only the essential elements that are being tested or controlled in the study.

While a good amount of the proposed data is theorized to come from the files of Village Reach, it is unclear whether all the proposed data will be available, and more information is needed to provide assurance that access to the data of interest will be available.

7. PLOS authors have the option to publish the peer review history of their article (what does this mean?). If published, this will include your full peer review and any attached files.

Reviewer #1: No

Reviewer #2: No

Reviewer #3: **Yes: **Peter Babigumira Ahabwe

Reviewer #4: No

Reviewer #5: No

Reviewer #6: No

Reviewer #7: No

---

## [Author Response · Author response to Decision Letter 0]

26 Jul 2024

We have attached a detailed 'Response for Reviewers' addressing comments raised.

---

## [Decision Letter · Decision Letter 1]

12 Sep 2024

Enhancing childhood immunization coverage in Mozambique and Malawi: study protocol of a mixed methods evaluation of the 'Let's talk about vaccines' multisite community-based participatory project

PONE-D-24-15369R1

Dear Linda,

We’re pleased to inform you that your manuscript has been judged scientifically suitable for publication and will be formally accepted for publication once it meets all outstanding technical requirements.

Kind regards,

Edison Arwanire Mworozi, M.D

Academic Editor

PLOS ONE

Thank you.

I have no further additions to make towards this paper. 

Good luck and thank you.

---

## [Editor Report · Acceptance letter]

17 Sep 2024

PONE-D-24-15369R1 

PLOS ONE

Dear Dr. Shuro, 

I'm pleased to inform you that your manuscript has been deemed suitable for publication in PLOS ONE. Congratulations! Your manuscript is now being handed over to our production team.

Kind regards, 

on behalf of

Dr. David Joseph Diemert 

Academic Editor

PLOS ONE